# New Antimicrobials for the Treatment of Neonatal Sepsis Caused by Multi-Drug-Resistant Bacteria: A Systematic Review

**DOI:** 10.3390/antibiotics12060956

**Published:** 2023-05-24

**Authors:** Chiara Poggi, Carlo Dani

**Affiliations:** 1Neonatal Intensive Care Unit, Department of Mother and Child Care, Careggi University Hospital, 50141 Florence, Italy; carlo.dani@unifi.it; 2Department of Neurosciences, Psychology, Drug Research and Child Health, University of Florence, 50141 Florence, Italy

**Keywords:** newborn, multi-drug resistance, antibiotics

## Abstract

Background: Infections by multi-drug-resistant (MDR) organisms are sharply increasing in newborns worldwide. In low and middle-income countries, a disproportionate amount of neonatal sepsis caused by MDR Gram negatives was recently reported. Newborns with infections by MDR organisms with limited treatment options may benefit from novel antimicrobials. Methods: We performed a literature search investigating the use in newborns, infants and children of novel antimicrobials for the treatment of MDR Gram negatives, namely ceftazidime/avibactam, ceftolozane/tazobactam, cefiderocol, meropenem/vaborbactam, imipenem/relebactam, and Gram positives with resistance of concern, namely ceftaroline and dalbavancin. PubMed, EMBASE, and Web of Science were searched. Results: A total of 50 records fulfilled the inclusion criteria. Most articles were case reports or case series, and ceftazidime/avibactam was the most studied agent. All studies showed favorable efficacy and safety profile in newborns and across different age cohorts. Conclusions: novel antibiotics may be considered in newborns for the treatment of MDR Gram negatives with limited treatment options and for Gram positives with resistance concerns. Further studies are needed to address their effectiveness and safety in newborns.

## 1. Introduction

Sepsis is among the leading causes of neonatal mortality and morbidity worldwide [1], accounting for about 1.3 (0.8–2.3 95% CI) million cases and 0.23 (0.179–0.276 95% CI) million neonatal deaths each year [2,3,4]. Antimicrobial resistance is globally spreading in the neonatal population [5,6,7], with particular concerns in low-income and middle-income countries (LMICs), while access to effective antimicrobials is still crucially limited in several regions [1].

Early-onset sepsis (EOS) is defined as a positive blood or cerebrospinal fluid culture taken within the first 72 h of life, while late-onset sepsis (LOS) occurs after the first 3 days of life [5,6]. Multi-drug resistance (MDR) bacteria were alarmingly reported in the latest years as causing agents of both EOS and LOS [6,7,8,9]. Moreover, several outbreaks caused by MDR organisms in Neonatal Intensive Care Units (NICUs) recently occurred in different regions, including high-income countries (HICs) [10,11,12,13,14], and consistent colonization of both patients admitted to NICUs [15,16,17] and of pregnant women [18] is reported, threatening the outcomes of both EOS and LOS cases.

### 1.1. Epidemiology

Among 0.68 million annual neonatal deaths associated with possible severe bacterial infection, an estimated 31% were attributable to resistant pathogens worldwide, with disproportionate risk in India, Pakistan, Nigeria, Congo, and China [1]. Studies from LMICs recently reported a higher amount of EOS in comparison HICs, accounting for 70–80% of all neonatal sepsis [2,9], and a high prevalence of Gram-negative strains, including MDR bacteria, in both EOS and LOS [2,6,8,19]. Gram negatives caused 39–64% of all neonatal sepsis [6,7,8] and the most frequently isolated organisms were *Klebsiella* spp., *S. marcescens, E. coli*, *Enterobacter,* and *A. baumannii* [6,8]. In recent studies, Gram-negative strains causing neonatal sepsis in LMICs across Africa and Asia were resistant to aminoglycosides in approximately 70% of cases [6,7], to cephalosporins in up to 84% of cases [5,6] and to carbapenems in 16–81% of cases [5,6,7]. In China, *E. coli* and *Klebsiella* spp. causing neonatal sepsis were reported as MDR organisms in 42 and 61% of cases, respectively [20,21], while carbapenem resistance was found in up to 31% of cases of LOS [21,22].

In HICs, *E. coli* is responsible for 35% of all EOS, and 51% of EOS in preterm infants [23], and Gram negatives account for 15–30% of LOS [24]. In the US, *E. coli* causing EOS displayed resistance to gentamicin in 10% of cases [25,26], while among all *E. coli* isolates from neonatal sepsis resistance to aminoglycosides was found in 14–17% of cases [25,27]. Carbapenem resistance was <5% [27]; however, 25% of all Gram negatives were resistant to at least one antimicrobial among gentamicin, piperacillin-tazobactam, 3rd–4th generation cephalosporins and carbapenems [26]. In the UK, 41% of *Enterobacter* spp. causing LOS showed resistance to the recommended combination of amoxicillin and cefotaxime and 18% to the combination of benzylpenicillin and gentamicin, while 15% of *E. coli* and 12% of *Klebsiella* spp. were resistant to both [28]. A study from Greece reported that *Klebsiella* spp. causing LOS were resistant to at least one aminoglycoside in almost half of cases [29]. In Germany, MDR organisms accounted for about 4% of LOS and 8% of EOS, with a predominance of extended-spectrum beta-lactamase (EBSL)-producing *E. coli* [30].

Despite routine screening for EBSL-producing and carbapenem-resistant *Enterobacterales* (CRE) is not performed in NICUs, colonization of admitted patients is increasingly documented worldwide [15,17,31,32]. Colonization by CRE at NICU admission was reported in 21–30% by studies from Vietnam and Turkey [15,31], and significant colonization acquisition during NICU stay was observed [15]. Colonization of 4% of admitted patients by MDR organisms was reported in a Belgian NICU [33] and in Italy, CRE were found to be significant colonizing agents of newborns admitted to different intensive care facilities [17,34].

Gram-positive strains, such as *Coagulase-negative Staphylococcus* (CONS)*, S. aureus,* and *Enterococcus* are at present the most frequent agents of nosocomial LOS in HICs [28,29] and vancomycin is among the most prescribed drugs in NICU [35]. Vancomycin-resistant *Enterococci* (VRE) are increasingly reported in neonatal sepsis worldwide [29,36,37] and accounted for 14% of all LOS in a network of Greek NICUs [29]. Resistance to glycopeptides of Gram-positive strains was reported in 13% of neonatal sepsis in India and 45% in Nigeria [5]. 

### 1.2. Current Treatment Options

To date, treatment options for MDR organisms in NICU are alarmingly limited [38,39,40], particularly for Gram-negative strains. Colistin is the main used antimicrobial for the treatment of MDR *P. aeruginosa, A. baumannii,* and CRE in newborns in the two last decades, with 75–100% of clinical success [41]. Meropenem at high doses or as an extended infusion or in association with other antimicrobials is the second most reported agent [40,41], while the use of tigecycline, fluoroquinolones, and polymyxin B is less frequently documented [24,40,42,43]. However, the polymyxins safety profile is not optimal, as nephrotoxicity has been reported in newborns in 10–19% of cases [42] along with significant electrolyte imbalances [42]. Treatment of Gram positives, such as methicillin-resistant *S. aureus* (MRSA), VRE, and CONS, with unfavorable susceptibility profiles or poor clinical response to oxacillin or vancomycin, is mainly based on linezolid and daptomycin [36,44]. However, linezolid showed variable clinical responses with clinical cure rates ranging from 63 to 100%, while the use of daptomycin might be questionable in the case of pneumonia [44].

Different antimicrobials are approved in adults for the treatment of infections caused by organisms with unfavorable susceptibility profiles. Particularly, beta-lactams/beta-lactamase inhibitors and cefiderocol are currently the cornerstones of the treatment of bloodstream infections (BSIs) and infections of different sites caused by Gram-negative strains with limited treatment options [45,46,47,48,49]. Ceftaroline and dalbavancin are among the main treatment options for Gram-positive strains with resistance concern [50,51], alongside with lipoglycopeptides such as televancin (not approved for use in EU) and oritavancin [52,53], the novel oxazolidinone tedizolid [54] and the 4th generation cephalosporin ceftobiprole [55]. Finally, eravacycline, a novel tetracycline with activity against Gram-positive cocci and Gram-negative bacilli, is considered for the treatment of intra-abdominal infections caused by susceptible strains [56]. 

Given the shortage of current treatment options in newborns, newly available antimicrobials such as beta-lactams/beta-lactamase inhibitors, namely ceftazidime/avibactam, ceftolozane/tazobactam, meropenem/vaborbactam, imipenem/relebactam, or cefiderocol may represent promising tools for the treatment of MDR Gram negatives in NICU, while ceftaroline and dalbavancin may represent treatment options for Gram-positive strains with resistances of concern. Despite the use of these antimicrobials has been increasingly reported in latest years in newborns and infants, to date, the possibility to extend their use to these populations has not been assessed. Therefore, this study aimed to review the current knowledge of the use of these antimicrobials in newborns.

## 2. Results

### 2.1. Study Selection 

Among a total of 986 records retrieved, 749 records were removed (736 duplicates); therefore, 237 records were screened and 196 were removed as they did not meet eligibility criteria. Therefore, a total of 68 articles were sought for retrieval and 18 articles were excluded as commentaries or narrative reviews not reporting original data or as studies documenting only antimicrobial susceptibility data. We decided not to include susceptibility studies as they did not provide data directly related to antimicrobials administration in clinical settings. Instead, we decided to include studies on pharmacokinetic simulation models, although not enrolling patients in clinical settings, as they were considered useful for the validation of administration schedules in the age groups of interest. Finally, 50 articles [57,58,59,60,61,62,63,64,65,66,67,68,69,70,71,72,73,74,75,76,77,78,79,80,81,82,83,84,85,86,87,88,89,90,91,92,93,94,95,96,97,98,99,100,101,102,103,104,105,106] were included in the present systematic review (Figure 1).

### 2.2. Treatment of MDR Gram-Negative Bacteria

We included 35 articles regarding the treatment of MDR Gram negatives with antimicrobials of interest, 16 on ceftazidime/avibactam [57,58,59,60,61,62,63,64,65,66,67,68,69,70,71,72], 12 on ceftolozane-tazobactam [72,73,74,75,76,77,78,79,80,81,82], 6 on cefiderocol [84,85,86,87,88,89,90], and 2 on meropenem-vaborbactam [90,91]. No eligible studies were retrieved for imipenem-relebactam. One study reported on both ceftazidime/avibactam and ceftolozane/tazobactam [72] and one [90] on both meropenem/vaborbactam and cefiderocol.

#### 2.2.1. Ceftazidime/Avibactam

Among 16 studies on ceftazidime/avibactam (Table 1), we included 12 case reports or case series, 2 randomized controlled trials (RCTs), and 2 pharmacokinetic studies. Case reports and case series included patients from birth to 16 years of age. In particular, 5/12 case reports or series [59,62,66,69,70] included newborns for a total of 11 newborns infected by carbapenem-resistant or extensively drug-resistant *K. pneumoniae*, 8 with LOS, 2 with LOS and meningitidis, and 1 with UTI, and 10/11 newborns achieved clinical cure; 4/5 studies on newborns included only preterm newborns [62,66,69,70], while 1 case series included both term and preterm newborns [59]. No significant drug-related AEs were reported; 2 mild AEs with an uncertain relationship with ceftazidime/avibactam were reported.

Included RCTs [60,61] enrolled patients ≥3 months old with urinary tract infections (UTIs) or complicated intra-abdominal infection, clinical cure achieved by ceftazidime/avibactam ranged from 95 to 98%.

Doses and administration schedules for ceftazidime/avibactam in newborns are at present extrapolated from pharmacokinetic data obtained from patients ≥3 months old [57,68].

#### 2.2.2. Ceftolozane/Tazobactam

Among 12 studies on ceftolozane/tazobactam (Table 2), we included 5 case reports or case series, 2 RCTs, and 5 pharmacokinetics studies. One RCT enrolled newborns with complicated UTIs [82] while the other one [83] enrolled patients >2 years of age with complicated intra-abdominal infections.

Case reports and case series and one retrospective study reported the use of ceftolozane/tazobactam in a total of 21 pediatric patients with infections of different sites (BSI, pneumonia, osteomyelitis, intra-abdominal infection) caused by MDR *P. aeruginosa* [72,73,75,76,79] with age ranging from 3 months to 18 years.

No severe drug-related AEs were reported, mild drug-related AEs included diarrhea, increased transaminases, and neutropenia [82].

Current doses and treatment regimens for ceftolozane/tazobactam were validated by pharmacokinetic studies specifically targeting newborns [74,77,78].

#### 2.2.3. Cefiderocol

Among the 6 included studies on cefiderocol (Table 3), 5 were case reports or series, and 1 was a pharmacokinetic study. In the case reports, 2/6 reported the use of cefiderocol in 2 preterm newborns with carbapenem-resistant *K. pneumoniae* [88,89], one with LOS [89], and one with LOS and necrotizing enterocolitis [88]. 

A pharmacokinetic simulation model validated doses and administration regimens for cefiderocol in newborns, providing specific doses basing on post-natal age and gestational age of the patients [84].

#### 2.2.4. Meropenem/Vaborbacatam

We identified 2 case reports on meropenem/vaborbactam regarding patients of 4 and 10 years [90,91] but no reports regarding newborns (Table 3).

### 2.3. Treatment of Gram-Positive Bacteria with Resistance of Concerns

We included 15 articles regarding the treatment of MDR Gram positives with antimicrobials of interest, 11 on ceftaroline [92,93,94,95,96,97,98,99,100,101,102] and 4 on dalbavancin [103,104,105,106].

#### 2.3.1. Ceftaroline

Among 11 studies on ceftaroline (Table 4), we included 3 RCTs, 1 clinical phase 2 trial, 3 case reports, 3 pharmacokinetics studies, and 1 retrospective study. Ceftaroline was studied in RCTs for the treatment of pneumonia and skin and skin structure infections (SSSIs) mostly caused by *S. aureus*, including MRSA, in patients ≥2 months of age [92,93,94], achieving clinical cure in 83–88% of cases [92,93,94]. Among case reports, 2/3 documented the use of ceftaroline in preterm newborns of 24–30 weeks of gestation infected by MRSA, one with LOS and pneumonia [96], and one with hepatic abscess and infected thrombus of the portal system [100].

Drug-related AEs were reported in 10–23% of treated patients [92,93]; the most frequently reported AEs were diarrhea, vomiting, dermatitis or rush, increased transaminases; two severe drug-related AEs were reported [92,93], one hypersensitivity event and one case of colitis by *C. difficilis* [94]. 

Pharmacokinetic studies specifically targeted to neonatal age [95,101,102] validated ceftaroline administration schedule of 8–10 mg/kg q8h [95,101] and demonstrated no differences in the probability of target attainment between 5 min or 60 min drug infusion [102]. 

#### 2.3.2. Dalbavancin

We included 4 studies on dalbavancin (Table 4), 3 pharmacokinetic studies, and one RCT. In the included RCT [106] dalbavancin was administered for the treatment of BSI in patients from birth to 3 months and SSSIs from birth to 18 years. 

Drug-related AEs were not reported.

## 3. Discussion

We identified 50 articles regarding the use of the antibiotics of interest, namely ceftazidime/avibactam, ceftolozane/tazobactam, cefiderocol, meropenem/vaborbactam, imipenem/relebactam, ceftaroline and dalbavancin in newborns, infants, and children. 

Most articles were case reports or case series or other retrospective studies, while few neonatal patients were enrolled in RCTs; the most studied antimicrobials were ceftazidime/avibactam and ceftolozane/tazobactam. The most frequently isolated MDR organisms in the included studies were *K. pneumoniae*, *P. aeruginosa,* and *E. coli* among Gram negatives, and MSSA, MRSA, and *Enterococcus* spp. among Gram positives. 

Ceftazidime/avibactam is a cephalosporin/beta-lactamase inhibitor, with excellent activity against KPC and OXA-48-like producing CRE and non-carbapenemase-producing CRE, and it is currently approved for use in patients ≥3 months for the treatment of complicated intra-abdominal infections, complicated UTIs, hospital-acquired pneumonia, and BSI associated with those conditions; it is also approved for treatment of infections caused by Gram negatives with limited treatment options [45]. Case reports on preterm infants [62,66,69,70] included newborns of 27–29 weeks, in whom treatment was started at 11–46 days of life and continued for 10 to 14 days in the case of UTI or BSI [66,70] and for 21 to 47 days in the case of meningitidis [62,69]. Among 11 patients treated with ceftazidime/avibactam 10 achieved microbiological and clinical cure, while one died. In 36 infants of 3 months–2 years of age with complicated UTIs, a clinical cure was observed in 99% of patients randomized to ceftazidime/avibactam vs. 90% of patients randomized to comparator cefepime [60]. Similar efficacy of ceftazidime/avibactam and meropenem was observed in 83 patients of 3 months–18 years of age with complicated intra-abdominal infections [61], indicating high efficacy of ceftazidime/avibactam in infants and children. Similarly, in adults, ceftazidime/avibactam showed higher efficacy in the treatment of infections caused by CRE in comparison with different combinations of colistin, tigecycline, fosfomycin, and carbapenems [43]. A recent meta-analysis [107] demonstrated that ceftazidime/avibactam was more effective than comparators to achieve clinical cure of infections caused by carbapenem-resistant *K. pneumoniae,* and patients treated with ceftazidime/avibactam also presented lower mortality rates at 28–30 days. Ceftazidime/avibactam was recently included in the treatment algorithm for both carbapenemase-positive and negative CRE infections in children, without restrictions of age [43]. It was also suggested to strongly consider the use of beta-lactams/beta-lactamase inhibitors, as ceftazidime/avibactam, imipenem/relebactam and meropenem/vaborbactam, for susceptible CRE isolates with MIC for meropenem ≥4 microg/mL or known to produce KPC based on rapid molecular diagnostic tests [43]. Meropenem/vaborbactam is at present approved for adult use in complicated UTIs, intra-abdominal infections, hospital-acquired pneumonia, or BSI associated with the previous conditions, and in general for infections by Gram negatives with limited treatment options [46]; similarly, imipenem/relebactam is approved for adult use [47]. At present no data are available for newborns.

Ceftolozane/tazobactam is a cephalosporin/beta-lactamase inhibitor with enhanced activity against *P. aeruginosa,* approved for the treatment of complicated intra-abdominal infection, and complicated UTIs including pyelonephritis with no restrictions of age, provided the patients are ≥ 7 days and ≥ 32 weeks of gestation, and for the treatment of hospital-acquired pneumonia in adults [48]. Among 95 patients with complicated UTIs, including 20 patients with age birth-3 months randomized to ceftolozane/tazobactam or meropenem, a clinical cure was observed in 94% vs. 100%, respectively [82], suggesting that ceftolozane/tazobactam is an effective treatment option in newborns, infants, and children. In patients >2 years with complicated intra-abdominal infections ceftolozane/tazobactam in combination with metronidazole was effective as meropenem and well-tolerated [83]. One case series [79] reported 3 infants with age 3–10 months with comorbidities treated with ceftolozane/tazobactam for pneumonia, 2 of whom were clinically cured. These results are consistent with data from adult patients, indicating the high efficacy of ceftolozane/tazobactam in patients with MDR Gram negatives infections, such as lower and upper UTIs [108] and intra-abdominal infections [109]. A recent meta-analysis [110] showed that ceftolozane/tazobactam was more effective in achieving clinical cure or microbiological eradication in comparison to polymyxin/aminoglycoside and quinolones in adults with Gram negatives infections, including MDR *P. aeruginosa.*


Cefiderocol is a siderophore cephalosporin with excellent activity against carbapenemases and it is currently approved for the treatment of Gram negatives with limited treatment options in adults [49]. Two case reports documented the successful treatment of preterm newborns of 27–31 weeks of gestation with LOS by VIM metallo-beta-lactamase producing *K. pneumoniae* [89] and LOS and necrotizing enterocolitis caused by KPC-producing *K. pneumoniae* [88] after the failure of netilmicin [89] and meropenem and colistin [88]. Treatment with cefiderocol was started at 9 [88] and 20 days of life [89] and continued for 14 and 9 days, respectively. In adults, cefiderocol showed similar efficacy to other available best comparators in the case of infections by MDR Gram negatives [111] and showed non-inferiority in comparison to meropenem for the treatment of hospital-acquired pneumonia [112]. Cefiderocol also showed superiority to the best available therapy and high-dose meropenem for the outcomes of clinical cure, microbiological eradication, and mortality at 28 days in the case of infections caused by metallo-beta-lactamase-producing Gram negatives [113].

Noticeably, all antimicrobial targeting MDR Gram negatives considered in this review were recently included in the guidelines of the Infectious Disease Society of America applying both to adult and pediatric patients, with no further age cohort specification [114], with an indication that all of them may be considered for the treatment of CRE, and, except for meropenem/vaborbactam, for the treatment of *P. aeuruginosa* with “difficult-to-treat resistance”. Cefiderocol is also recommended for the treatment of carbapenem-resistant *A. baumannii* [114].

Ceftaroline is a 5th generation cephalosporin with activity against Gram positives including MRSA and MDR *S. pneumoniae*, and it is currently approved for the treatment of patients of any age, including newborns, with SSSIs or community-acquired pneumonia [50]. In 11 infants of 7–60 days with LOS treated with ceftaroline plus ampicillin and optional aminoglycoside, no treatment failure was observed [98]. In patients ≥2 months with complicated community-acquired pneumonia, ceftaroline showed similar efficacy in comparison to vancomycin plus ceftriaxone, with clinical cure observed in 83% vs. 78% of cases, respectively [93]. Ceftaroline also showed similar efficacy in comparison to ceftriaxone for the treatment of community-acquired pneumonia, with clinical cure observed in 92% vs. 89% of cases, respectively [92]. Likewise, ceftaroline was proved highly effective for the treatment of adult pneumonia, and a recent meta-analysis found a higher probability of clinical cure with ceftaroline in comparison to ceftriaxone [115]. Moreover, ceftaroline recently showed non-inferiority in comparison to daptomycin for the treatment of BSIs caused by MRSA without pulmonary origin [116]. For the treatment of 159 patients ≥2 months of age with SSSIs ceftaroline achieved clinical cure in 96% vs. 88% of comparators, vancomycin or cefazolin [94], in accordance with data from adult patients [117,118,119]. In 2 case reports, ceftaroline was effective in preterm newborns of 24–30 weeks of gestation with BSI by MRSA, one with LOS and pneumonia [96], and one with hepatic abscess and infected thrombus of the portal system [100]. In one case [96], treatment with ceftarolin was started at 43 days of life, after failure of oxacillin, vancomycin, and rifampin, and administered for 21 days [96], while in the other case, treatment was started at 54 days of life, after failure of vancomycin, daptomycin, and linezolid, and continued for 18 days [100]. Successful pharmacokinetic target attainment was also reported with the administration of 8.5 mg/kg q8h [96].

Dalbavancin is a long-acting semisynthetic lipoglycopeptide antibiotic with bactericidal activity against Gram-positive pathogens, including *S. aureus* including MRSA, *S. pneumoniae, S. agalactiae, S. pyogenes,* and *Enterococcus* spp., and it is currently approved for the treatment of SSSIs in patients >3 months [51]. Dalbavancin was administered for the treatment of BSIs in patients from birth to 3 months and SSSIs from birth to 18 years known or suspected to be caused by susceptible Gram positives [106], mainly MSSA; 5 patients in the cohort birth-3 months, including 3 patients with age < 1 month, were treated with single dose dalbavancin, with excellent overall efficacy.

At present, the susceptibility of pediatric and neonatal isolates to novel antimicrobials is excellent in HICs. In studies on pediatric and neonatal isolates, *Enterobacterales* showed excellent susceptibility of 97–100% to ceftazidime/avibactam [120,121], while *P. aeruginosa* maintained high susceptibility of 96–100% in the general pediatric population including newborns [121] but showed poor susceptibility of 47% in colonized pediatric patients with cystic fibrosis [122]. At present overall susceptibility of pediatric Gram negatives isolates to ceftolozane/tazobactam is excellent in HICs [123,124,125]; however, slightly lower susceptibility has been shown for *K. pneumoniae* [125] and resistance to ceftolozane/tazobactam was reported in approximately half of the cases for pediatric cystic fibrosis patients colonized with MDR *P. aeruginosa* [122]. In 1460 isolates from pediatric respiratory tract infections and SSSIs, including 263 isolates obtained from patients <1 year, susceptibility to ceftaroline was 100% for *H. influenzae, S. aureus,* and *E. coli,* 99.6% for *S. pneumoniae,* and 97% for *Klebsiella* spp. [126].

No significant safety issues in newborns and infants emerged from the included studies. For the treatment of Gram negatives, the use of ceftazidime/avibactam was not associated with any drug-related AEs, in accordance with the re-assuring safety profile observed in adults [43,107]. In preterm newborns 2 AEs were reported to have an uncertain association with the drug, one case of thrombocytopenia not requiring transfusion [62], and one case of transient glycosuria [66]. Drug-related adverse events were reported for ceftolozane/tazobactam by 2 RCTs and included diarrhea, increased transaminases, and dermatitis or rush, similarly to AEs displayed by adult patients [82,83]. No AEs were reported for preterm newborns. Cefiderocol and meropenem/vaborbactam we not associated with any drug-related AEs. Mild drug-related AEs were reported in 10–23% of patients treated with ceftaroline, including diarrhea, vomiting, dermatitis or rush, and increased transaminases. Two severe ceftaroline-related AEs, one hypersensitivity event and one case of colitis by *C. difficilis* were observed, both beyond the neonatal period [94], indicating similar safety profile to adult patients [92,93].

This review has some limitations. First, a modest amount of data results from RCTs or clinical studies properly designed to assess the efficacy and safety of new antimicrobials in infants and newborns, while most of the included studies were case reports or case series. Second, included studies presented variable study designs, not allowing direct comparison of the results. Third, most RCTs enrolled infants ≥2 or 3 months of age, thus slightly beyond the neonatal period. However, newborns and particularly preterm newborns may significantly differ from slightly older patients in terms of pharmacokinetic variables [127]. Newborns, particularly if preterm, show a higher volume of distribution of antimicrobial drugs, but lower renal drug clearance, resulting in higher loading dose but lower maintenance dose of the drug [127], with poorly predictable effects on efficacy and toxicity [127]. Finally, most data on the use of novel antimicrobials in newborns were obtained from studies performed in HICs. However, patients who may benefit from new antimicrobials may partially differ across countries, as a particularly high rate of MDR organisms were found in EOS in term or mild preterm newborns in LMICs, while infections with MDR organisms in HICs usually occur in very preterm newborns with healthcare-associated LOS. 

Overall, available data indicate that novel antimicrobials against MDR Gram-positive and Gram-negative organisms are effective and safe in the pediatric and neonatal population, and therefore, they can be considered a useful treatment in case of infections caused by MDR organisms in NICU, when other treatment options are limited or absent. However, data on the use of these antimicrobials are still limited for children and newborns; therefore, properly designed RCTs in these populations are warranted, including newborns and preterm newborns, to specifically assess efficacy and safety in these age groups. Finally, limited or no availability of novel antimicrobials in LMICs might represent a significant issue, as those areas would likely mostly benefit from novel agents in consideration of the high rates of MDR Gram negatives reported. 

## 4. Materials and Methods

### 4.1. Search Strategy

This systematic review was carried out according to the Preferred Reporting Items for Systematic Reviews and Meta-Analyses (PRISMA) guidelines (Figure 1). A literature search was conducted on 25 March 2023, using the following databases: PubMed Medline, EMBASE, and Web of Science. 

The search strategy included the following terms and was performed for each considered antibiotic (ceftazidime/avibactam, ceftolozane/tazobactam, cefiderocol, meropenem/vaborbactam, imipenem/relebactam, ceftaroline, dalbavancin): “antibiotic of interest” AND “neonate” OR “newborn” OR “neonat” OR “infant” OR “child” OR “pediatric”. No date restriction was applied. The literature search was limited to the English language. Articles were checked for duplication.

### 4.2. Eligibility

Two reviewers (C.P., C.D.) independently assessed eligibility. Titles and abstracts of all retrieved articles were screened to identify potentially eligible studies, and all selected articles were analyzed in full text for conclusive evaluation. Eligibility criteria for the present study were as follows: (1) studies investigating the use of new antimicrobials of interest (2) in newborns, infants, and children (3) with BSIs or infection of any site, or (4) pharmacokinetics studies in the same population. Based on the extremely recent introduction of novel antimicrobials in newborns and pediatric age in general, we decided to consider all studies on patients younger than 18 years, to collect all available data for the developmental age. The eligible study design included RCTs, retrospective studies, case reports/series, and pharmacokinetic studies. Reviews, commentaries, or meta-analyses were considered not eligible. Considered outcomes were clinical cure, microbiological eradication, safety issues, and pharmacokinetic target attainment or dose validation, depending on different types of considered studies.

## 5. Conclusions

In conclusion, robust evidence on the efficacy and effectiveness of novel antimicrobials for the treatment of MDR Gram positives and Gram negatives is lacking. However, all available data suggest high effectiveness and favorable safety profile of the considered novel antimicrobials in the neonatal population, including preterm newborns. Therefore, these drugs might be regarded as useful treatments in newborns and infants with infections caused by MDR organisms with limited treatment options. Further studies are warranted to specifically address indications and safety profiles in infants and newborns.

## Figures and Tables

**Figure 1 antibiotics-12-00956-f001:**
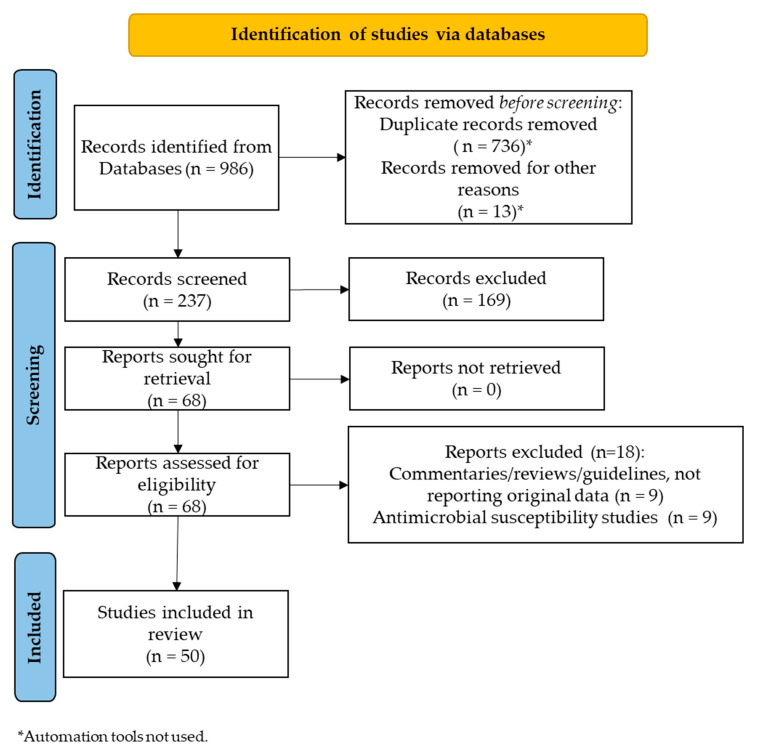
PRISMA 2020 diagram for study selection.

**Table 1 antibiotics-12-00956-t001:** Studies on ceftazidime/avibactam.

	Study Type	Country	Patient Characteristics	N	Organisms	Study Intervention	Outcomes
Bradley 2016 [57]	Phase I Pharmacokinetics	US	3 month–2 years	8	n.p.	50 mg/kg Single 2 h infusion	Pharmacokinetics model validation 1/8 drug-related mild AEs (sinus tachycardia)
Tamma 2018 [58]	Case report	US	2 month BSI	1	*Burkholderia cepacia*	50 mg/kg q8h Continuous infusion Duration: 6 weeks	1/1 BC sterilization 1/1 clinical recovery No drug-related AEs
Iosifidis 2019 [59]	Case series	Greece	Newborns including preterm, LOS	6	XDR or PDR *Klebsiella pneumoniae*	50 mg/kg q8h Duration: 4–38 days (median 14 d)	6/6 BC sterilization 6/6 clinical recovery No drug-related AEs
Bradley 2019 [60]	Phase II RCT	US	3 month–2 years UTI	95	*Enterobacterales*	Randomization 3:1 to C/A 40–50 mg/kg q8h or cefepime Duration: ≥72 h	In C/A group: 17/22 urine sterilization 21/22 clinical recovery No drug-related AEs
Bradley 2019 [61]	Phase II RCT	US	3 month–18 years complicated intra-abdominal infection	83	*Escherichia coli* *Pseudomonas aeruginosa*	Randomization 3:1 to C/A 40–50 mg/kg q8h + metronidazole or MEM Duration: ≥72 h	In C/A group: 57/61 clinical recovery No drug-related AEs
Esposito 2019 [62]	Case report	Italy	Preterm infant BW 680 g, LOS + meningitidis (30 DOL)	1	KPC-producing *Klebsiella pneumoniae*	75 mg/kg q8h Duration: 47 days	1/1 BC sterilization 1/1 clinical recovery 1/1 uncertain mild drug-related AE (thrombocytopenia)
Vargas 2019 [63]	Case report	Italy	14 years BSI and pneumonia	1	MDR *Klebsiella pneumoniae*	2.5 g q8h Duration: 14 days	1/1 Clinical cure No drug-related AEs
Nguyem 2019 [64]	Case report	US	16 years CF Pulmonary exacerbation	1	*Burkholderia cepacia*	2.5 g q8h Duration: 14 days	1/1 Sputum sterilization 1/1 Clinical cure No drug-related AEs
Hobson 2019 [65]	Case report	France	3 years BSI in acute leukemia	1	MDR *Morganella morganii*	150 mg/kg/d Duration: n.p.	1/1 BC sterilization 1/1 Clinical cure No drug-related AEs
Coskum 2020 [66]	Case report	Turkey	Preterm infant GA 27 weeks, UTI (25 DOL)	1	PDR *Klebsiella pneumoniae*	40 mg/kg q8h Duration: 10 days	1/1 urine sterilization 1/1 clinical recovery 1/1 uncertain mild drug-related AE (glycosuria)
Ji 2021 [67]	Case report	China	2 month Shoulder osteomyelitis	1	CR *Klebsiella pneumoniae*	200 mg q8h Duration: 14 days	1/1 clinical recovery No drug-related AEs
Franzese 2021 [68]	Pharmacokinetics	US	3 month–18 years	153	n.p.	40 mg/kg for infants < 6 month Single 2 h infusion	Pharmacokinetic model validation
Asfour 2022 [69]	Case series	Saudi Arabia	Preterm infants GA 27–28 weeks LOS + meningitidis (DOL 11) LOS (DOL 37)	2	CR *Klebsiella pneumoniae*	50 mg/kg q8h Duration: 5–21 days	2/2 BC sterilization 1/2 clinical recovery 1/2 death No drug-related AEs
Nascimento 2022 [70]	Case report	Brazil	Preterm infant GA 29 weeks LOS (46 DOL)	1	MDR *Klebsiella pneumoniae*	40 mg/kg q8h Duration: 14 days	1/1 BC sterilization 1/1 clinical recovery No drug-related AEs
Almangour 2022 [71]	Case report	Saudi Arabia	2 years Ventriculoperitoneal shunt infection	1	MDR *Pseudomonas aeruginosa*	62.5 mg/kg q8h Duration: 21 days	1/1 CSF sterilization 1/1 Clinical cure No drug-related AEs
Perruccio 2022 [72]	Case series	Italy	7 month–17 years Malignancy BSI, pneumonia, appendicitis	21 (+4 C/T)	MDR *Enterobacterales*	50mg/kg q8h Duration: 6–19 days	23/25 Clinical cure 2/25 Death No drug-related AEs

AEs: adverse events, BC: blood culture, BSI: bloodstream infection, CSF: cerebrospinal fluid, CF: cystic fibrosis, C/A: ceftazidime/avibactam, C/T: ceftolozane/tazobactam, CR: carbapenem-resistant, DOL: days of life, GA: gestational age, LOS: late-onset sepsis, MDR: multi-drug resistant, MEM: meropenem, PDR: pan-drug resistant, UTI: urinary tract infection.

**Table 2 antibiotics-12-00956-t002:** Studies on ceftolozane/tazobactam.

	Study Type	Country	Patient Characteristics	N	Organisms	Study Intervention	Outcomes
Aitken 2016 [73]	Case report	US	9 years BSI in acute leukemia	1	MDR *P. aeruginosa*	1st course: 50 mg/kg q8h Duration: 3 weeks 2nd course: 40 mg 7 Kg q6h Duration: 3 weeks	1/1 BC sterilization (relapse after 1st course) 1/1 Clinical cure (relapse after 1st course) No drug-related AEs
Bradley 2018 [74]	Phase I Pharmacokinetics	US	>7 days <18 years	34	Proven or suspected Gram neg infection	20 mg/kg for pts 7d–3 month Single 1 h infusion	Pharmacokinetics model validation No drug-related AEs
Martin-Cazana 2019 [75]	Case report	Spain	5 years BSI and endocarditis in congenital heart disease	1	MDR *P. aeruginosa*	50 mg/kg q8h Extended infusion (3 h) Duration: 6 weeks	1/1 BC sterilization 1/1 Clinical cure No drug-related AEs
Zikri 2019 [76]	Case report	Saudi Arabia	14 years BSI and pneumonia in immunodeficiency	1	MDR *P. aeruginosa*	1.5 g q8h Duration: n.p.	1/1 Clinical cure No drug-related AEs
Ang 2019 [77]	Phase I Pharmacokinetics	US	>7 days <3 month	13	Proven or suspected Gram neg infection	20 mg/kg Single 1 h infusion	Pharmacokinetics model validation No drug-related AEs
Larson 2020 [78]	Pharmacokinetics	US	simulation model (0–18 years)	—	—	—	Recommended doses: 20 mg/kg q8h (birth–12 years)
Molloy 2020 [79]	Case series	US	3 month–19 years	13	MDR *P. aeruginosa* (7 pneumonia, 3 CF, 2 abdominal infections, 1 osteomyelitis)	20 mg/kg q8h Duration: up to 8 weeks	12/13 clinical cure 2/13 uncertain drug-related AEs (transaminitis, neutropenia) No drug-related AEs in pts < 1 years
Arrieta 2020 [80]	Pharmacokinetics	US	2–18 years CF Respiratory disease	18	n.p.	18–30 mg/kg (2–7 years)	100% target attainment probability No differences CF vs. non-CF
Butragueno-Laiseca 2020 [81]	Pharmacokinetics	US	9–19 month CF	3	MDR *P. aeruginosa*	30–40 mg/kg q8h	Recommended doses: 35 mg/kg q8h if normal renal function 10 mg/kg q8h if acute renal injury 30 mg/kg if renal replacement therapy
Perruccio 2022 [72]	Case series	Italy	7 month–17 years Malignancy BSI, pneumonia, appendicitis	4 (+21C/A)	MDR *Enterobacterales*	1g q8h Duration: 14–20 days	23/25 Clinical cure 2/25 Death No drug-related AEs
Roiledes 2023 [82]	Phase II RCT	US/Europe	7d–18 years Complicated UTI	95	*E. coli* *K. pneumoniae* *P. aeruginosa*	Randomization 3:1 to C/T 20 mg/kg q8h or MEM Duration: 7–14 days	94% vs. 80% clinical cure 14% mild drug-related AEs (diarrhea, increased transaminases, neutropenia) No severe drug-related AEs
Jackson 2023 [83]	Phase II RCT	US/Europe	0–18 years Complicated intra-abdominal infection	91	*E. coli*	Randomization to C/T 20 mg/kg q8h + metronidazole or MEM	Clinical cure: 80 vs. 95% In C/T group: 13/70 mild drug-related AEs (diarrhea, increased transaminases, increased alkaline phosphatase, vaginal mycosis, dysgeusia) No sere drug-related AEs

AEs: adverse events, BC: blood culture, BSI: bloodstream infection, CF: cystic fibrosis, C/T: ceftolozane/tazobactam, DOL: days of life, GA: gestational age, MDR: multi drug resistant, MEM: meropenem, UTI: urinary tract infection.

**Table 3 antibiotics-12-00956-t003:** Studies on cefiderocol and meropenem/vaborbactam.

	Antibiotic	Study Type	Country	Patient Characteristics	N	Organisms	Study Intervention	Outcomes
Katsube 2019 [84]	Cefiderocol	Pharmacokinetics	Japan	simulation model (0–18 years)	—	—	—	Recommended doses: GA < 32 weeks: <2 month 30 mg/kg q8h >2 month 40 mg/kg q8h GA ≥ 32 weeks: <2 month 40 mg/kg q8h >2 month 60 mg/kg q8h
Alamarat 2020 [85]	Cefiderocol	Case report	US	15 years Chronic osteomyelitis	1	XDR *P. aeruginosa* + ESBL producing *K. pneumoniae*	2g q8h Extended infusion (3 h) Duration: 14 weeks (+surgery)	1/1 Bone biopsy specimen sterilization 1/1 Clinical cure No drug-related AEs
Warner 2021 [86]	Cefiderocol	Case series	US	0–18 years CF Pulmonary exacerbation	2	*Achromobacter xylosoxidans*	60 mg/kg q8h	2/2 clinical recovery 1/2 relapse No drug-related AEs
Grasa 2021 [87]	Cefiderocol	Case report	Spain	2 years BSI in Burkitt lymphoma	1	Carbapenemase-producing *P. aeruginosa*	60 mg/kg q8h Duration: 7 days	1/1 Clinical cure No drug-related AEs
Bawankule 2022 [88]	Cefiderocol	Case report	India	Preterm 27 weeks GA DOL 9 LOS + NEC	1	KPC-producing *K. pneumoniae*	30 mg/kg q6h Duration: 14 days	1/1 BC sterilization 1/1 clinical recovery No drug-related AEs
Monari 2023 [89]	Cefiderocol	Case report	Italy	Preterm 31 weeks GA DOL 20 LOS	1	KPC-producing *K. pneumoniae*	60 mg/kg loading dose 40 mg/kg q8h Extended infusion (3–4 h) Duration: 9 days	1/1 BC sterilization 1/1 clinical recovery No drug-related AEs
Henretty 2018 [90]	Meropenem/Vaborvactam	Case report Pharmacokinetics	US	4 years CLABSI	1	KPC-producing *K. pneumoniae*	40 mg/kg q6h, 3 h infusion Duration: 14 days	1/1 BC sterilization 1/1 Clinical cure 100% MEM concentration > MIC
Gainey 2020 [91]	Meropenem/Vaborvactam +Cefiderocol +bacteriophage	Case report	US	10 years CF	1	*Achromobacter* spp. resistant to FDC and M/V	Duration: 14 days	1/1 Sputum sterilization 1/1 Clinical cure

AEs: adverse events, CLABSI: central line-associated bloodstream infection, CF: cystic fibrosis, DOL: days of life, ESBL: extended spectrum beta-lactamase, KPC: *Klebsiella pneumoniae* carbapenemase; FDC: cefiderocol, GA: gestational age, M/V: meropenem/vaborbactam, NEC: necrotizing enterocolitis, UTI: urinary tract infection, XDR: extensively drug-resistant.

**Table 4 antibiotics-12-00956-t004:** Studies on ceftaroline and dalbavancin.

	Antibiotic	Study Type	Country	Patient Characteristics	N	Organisms	Study Intervention	Outcomes
Cannavino 2016 [92]	Ceftaroline	RCT	US, Europe	2 month–17 years, CABP	160		8 mg/kg q8h (2–6 month) 1 h infusion Duration:	92% clinical cure 10% drug-related AEs (diarrhea, vomiting)
Blumer 2016 [93]	Ceftaroline	RCT	US	2 month–18 years Complicated CABP	38 2 month–2 years: 6	3/29 MSSA 1/29 MRSA Others: *S. pneumonia, S. pyogenes, H. Influenzae, P. aeruginosa*	Randomization 3:1 to ceftaroline 10 mg/kg q8h (2–6 month) or ceftriaxone+ vancomycin Duration: 3–19 days	In ceftaroline group: 24/29 clinical cure 7/30 drug-related AEs (vomiting, diarrhea, increased AST/ALT, dermatitis, rush)
Korczowski 2016 [94]	Ceftaroline	RCT	US, Europe, South America, Africa	2 month–17 years SSSI	159	Mostly *S. aureus* (42% MRSA)	Randomization 2:1 to ceftaroline 8 mg/kg q8h (2–6 month) or comparator (cefazolin or vancomycin) Duration: 5–14 days	96% clinical cure 94% microbiological eradication MRSA 89% microbiological eradication
Riccobene 2017 [95]	Ceftaroline	Pharmacokinetics	US	— simulation model (0–18 years)	—	—	—	8 mg/kg q8h (2 month–2 years) has >97% probability of target attainment
Salerno 2018 [96]	Ceftaroline	Case report	US	Preterm 24 weeks GA DOL 43 LOS and pneumonia	1	MRSA	8.5 mg/kg q8h Duration: 21 days	1/1 BC sterilization 1/1 clinical cure No drug-related AEs Pharmacokinetics target attained
Branstetter 2020 [97]	Ceftaroline	Retrospective	US	0–21 years CF pulmonary exacerbation	90*	71/90 MRSA 21/90 *Pseudomonas* coinfection	Randomization 1:1 to ceftaroline or vancomycin Doses and duration not reported	No differences in lung function and readmission rate
Bradley 2020 [98]	Ceftaroline	Phase II	US	7–60 d, LOS	11	*E. coli* *Staphyloccocus* spp.	4–6 mg/kg q8h, 1h infusion Duration: 2–14 days	0/11 clinical failure 1/11 mild AE (diarrhea) Pharmacokinetic target attainment probability > 95%
Ferguson 2020 [99]	Ceftaroline	Case report		20 month BSI, wound infection, endocarditis, septic pulmonary emboli	1	MRSA	8 mg/kg q6h Duration: 1 days (shifted to vancomycin)	Resistance to ceftaroline despite no previous exposure
Heger 2022 [100]	Ceftaroline	Case report	US	Preterm GA 30 weeks DOL 54 Hepatic abscess and infected portal thrombus	1	MRSA	8 mg/kg q8h Duration: 18 days	1/1 clinical cure No drug-related AEs
Chan 2021 [101]	Ceftaroline	Pharmacokinetics	US	simulation model (0–18 years)	—	—	—	10 mg/kg q8h (2 month–2 years) has 99% probability of target attainment
Riccobene 2021 [102]	Ceftaroline	Pharmacokinetics	US	simulation model 2 month–18 years	—	—	—	5 min or 60 min infusion have >99% probability of target attainment
Bradley 2015 [103]	Dalbavancin	Pharmacokinetics	US	12–17 years	10	n.p.	15 mg/kg single dose	Slightly lower exposure than adults given 1 g
Gonzalez 2017 [104]	Dalbavancin	Phase I Pharmacokinetics	US	3 month–11 years Suspected or confirmed bacterial infection	43	n.p.	—	Recommended regimens 3 month–6 years: 15 mg/kg day 1 + 7.5 mg/kg day 8 OR 22.5 mg/kg day 1 5 probable drug-related AEs (rash, dermatitis, urticaria, elevated liver enzymes); no drug-related severe AEs
Carrothers 2023 [105]	Dalbavancin	Pharmacokinetics	US	— simulation model 0–18 years SSSI, neonatal sepsis	211	n.p.	—	22.5 mg/kg 30 min single infusion has probability of target attainment >94%
Giorgobiani 2023 [106]	Dalbavancin	Phase III RCT	US	0–18 years SSSI 0–3 month BSI	191 (0–3 month: 5)	*S. aureus, S. pyogenes, S. mitis/S. oralis, E. faecalis*	Randomization 3:3:1 to DAL 1 dose or DAL 2 doses or comparator (<3 month: DAL 1 dose 18 mg/kg or comparator) 30 min infusion Duration	Clinical cure 97 vs. 99 vs. 89% No drug-related AEs

BSI: bllodstream infection, CABP: community-acquired bacterial pneumonia, CF: cystic fibrosis, CARTI: community-acquired respiratory infection, DOL: days of life, GA: gestational age, LOS: late-onset sepsis, MRSA: methicillin resistant *S. aureus*, MSSA: methicillin sensitive *S. aureus*, SSSI: skin and skin structure infection.

## Data Availability

Data sharing not applicable.

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
