# Peer review of "New Antimicrobials for the Treatment of Neonatal Sepsis Caused by Multi-Drug-Resistant Bacteria: A Systematic Review"

_antibiotics, 2023, doi:10.3390/antibiotics12060956_

Round 1

Reviewer 1 Report

In this systematic review by Poggi and Dani, the authors have reviewed the existing literature to determine the usage of several antimicrobials and their efficacy in the light of increasing emergence of multidrug resistant (MDR) pathogens amongst newborns and infants. Based on this existing knowledge, they proposed that several antimicrobial combinations like ceftazidime/ avibactam, ceftolozane/ tazobactam, cefiderocol, meropenem/ vaborbactam, imipenem/ relebactam should be further investigated and possibly considered for treatment of newborns and infants, particularly those infected with MDR strains. The study is overall well-conducted. However, there are a few comments/concerns that should be addressed before further consideration.

1.     Throughout the review, the authors have described the use of different antimicrobial combinations like ceftazidime/ avibactam, ceftolozane/ tazobactam, cefiderocol, meropenem/ vaborbactam, imipenem/ relebactam as “new or novel antimicrobial”. However, it is not really the use of new antibiotics or antimicrobials. These are used in treatments of the adult population. In the current study, the authors are actually trying to determine if these antimicrobials could also be extended to treat the newborn or infant patient populations. It is therefore the new use of these antibiotics in the neonatal population.

2.     The authors discussed the usage and efficacy of different antimicrobials in pediatric sepsis, both early and late onsets. It will be helpful for the readers to be reminded of these two clinical situations and what the distinguishing criteria are in the introduction section.

3.     It is not clear why the authors excluded the susceptibility studies (n=9) as described in figure 1. Also, it is unclear why the authors included the studies based solely on simulation models (Ref 66, 72) in their main analysis. Their rationale for including these studies should be specified.

4.     Figure 1: the authors mention identifying the studies via databases and registers. What were the registers used for the same? All the figures will benefit from a detailed figure legend describing the abbreviations used. What does (**) stand for in Figure 1?

5.     The authors should do an overall review for correct scientific notations, typos, language, spelling and grammar in the entire text. The species names should start with lowercase.

6.     Some of the references are incorrect. For example, line 58, 290, 347.

The authors should do an overall review for correct scientific notations, typos ( for example in the tables), language, spelling and grammar in the entire text. The species names should start with lowercase. Some of the sentences are incomprehensible. It is suggested that the authors consider revising the elaborate sentences, for example - line 309 and 310, line 323, line 332, opening sentence in abstract. 

Author Response

Dear reviewer,

We really appreciated the time and effort that you dedicated to review our manuscript. All you comments and suggestion strongly helped to improve our paper. We hope that you could give further consideration to the revised version of manuscript.

Please find below our point by point response to your comments.

  1. Thanks for your comment, which gave us the opportunity to clarify the importance and the scope of our review. The fact that the considered antimicrobials are currently used in adults and that our study explores the possibility too extend their use in newborns and infants was addressed in the revised version of the manuscript (lines 97-116). Moreover the terms “novel/new antibiotics/antimicrobials” were changed into “antibiotics/antimicrobials of interest” (lines 192, 250, 281).
  2. According to your recommendation, we added the definitions of early and late-onset sepsis in newborns (lines 33-35).
  3. We decided to include in this systematic review studies with direct implication for antimicrobial use and administration in the neonatal population. Therefore, susceptibility studies were excluded as not directly related to the use of the antimicrobials in clinical practice, however susceptibility studies were taken into account in the Discussion section, as they provide relevant data on the potential feasibility of the use of these drugs in newborns and pediatrics. On the other hand, along with clinical studies or case report/series, we decided to include pharmacokinetics studies or models assessing or validating dosing regimens for newborns and infants, as administration schedules for these populations are often adjusted from adult ones, without investigations of age-specific models. Following your comment, these aspects were specified in the revised version of the manuscript (lines 149-153).
  4. Registers were not used, as specified in Material and methods section literature search was performed in PubMed, EMBASE and Web of Science. Therefore, the title of Fig.1 was corrected (line 158). “**” was removed from Figure 1 (line168).
  5. We performed the corrections needed throughout the text.
  6. References were checked and corrected if needed.
  7. Elaborate sentences were fixed throughout the manuscript (lines 390, 403, 412) and the opening sentence of abstract was changed (lines 10-12).

Reviewer 2 Report

This study did a comprehensive review of novel antibiotic in the treatment of sepsis among neonate. Overall, the topic is important and the manuscript is well-written. Thus, I just have several minor suggestions as the following.

1.     The present title did not fit well the article and need revision to be consistent with the text.

2.     In the introduction, please add more description about other new antibiotics, such as other lipoglycopeptide, tedizolid, new tetracycline – eravacycline, and ceftobiprole.

3.     Please move the part of method and material to the site before result section.

4.     In addition, please define which novel antibiotic was the antibiotic of interest in the literature search.

Author Response

Dear reviewer,

We would like to thank you for the time and effort that you dedicated to review our manuscript.  We really appreciated your suggestions and recommendations, which were really helpful to improve our manuscript. We hope you could further consider the revised version of the manuscript for publication.

Please find below our point by point response to your comments.

  1. According to your suggestion the title was modified as follows: “New antimicrobials for the treatment of neonatal sepsis caused by multi-drug resistant bacteria: a systematic review” (lines 2-3).
  2. According to your recommendation, a brief paragraph on other novel antimicrobials used in adults was added in the introduction section (lines 97-107).
  3. Material and methods section was moved before Results section (line 117).
  4. The antibiotics of interest which were included in the literature search were specified (lines 124-125).

Reviewer 3 Report

I thank the authors for the opportunity to read this interesting paper and I congratulate them for the work they have done.

Author Response

Dear reviewer,

We would like to thank you for the time and effort that you dedicated to our manuscript. We really appreciate you recognition to our work.